# Genotype Prevalence of Lactose Deficiency, Vitamin D Deficiency, and the Vitamin D Receptor in a Chilean Inflammatory Bowel Disease Cohort: Insights from an Observational Study

**DOI:** 10.3390/ijms241914866

**Published:** 2023-10-03

**Authors:** Tamara Pérez-Jeldres, M. Leonor Bustamante, Roberto Segovia-Melero, Nataly Aguilar, Fabien Magne, Gabriel Ascui, Denisse Uribe, Lorena Azócar, Cristián Hernández-Rocha, Ricardo Estela, Verónica Silva, Andrés De La Vega, Elizabeth Arriagada, Mauricio Gonzalez, Gian-Franco Onetto, Sergio Escobar, Pablo Baez, Alejandra Zazueta, Carolina Pavez-Ovalle, Juan Francisco Miquel, Manuel Álvarez-Lobos

**Affiliations:** 1Department of Gastroenterology, School of Medicine, Pontificia Universidad Católica de Chile, Santiago 8320000, Chile; 2Department of Gastroenterology, Hospital San Borja Arriarán, Santiago 8360160, Chilema64go@hotmail.com (M.G.);; 3Biomedical Sciences Institute, Faculty of Medicine, Universidad de Chile, Santiago 8380453, Chile; 4Fundación Diagnosis, Santiago 7500580, Chile; 5Red Salud Arauco, Santiago 7560994, Chile; 6La Jolla Institute for Immunology, San Diego, CA 92037, USA; 7Instituto de Nutrición y Tecnología de Alimentos, Facultad de Medicina Universidad de Chile, Santiago 8380453, Chile; 8Center of Medical Informatics and Telemedicine, University of Chile, Santiago 8380453, Chile

**Keywords:** single-nucleotide polymorphism, Latin American, inflammatory bowel disease, vitamin D, lactose intolerance

## Abstract

Lactose intolerance (LI) and vitamin D deficiency (VDD) have been linked to inflammatory bowel disease (IBD). We conducted an observational study in 192 Chilean IBD patients to investigate the prevalence of a specific gene variant (LCT-13910 CC genotype) associated with LI and the prevalence of VDD/Vitamin D Receptor (VDR) gene variants. Blood samples were analyzed using Illumina’s Infinium Global Screening Array. The LCT-13910 CC genotype was found in 61% of IBD patients, similar to Chilean Hispanic controls and lower than Chilean Amerindian controls. The frequency of the LCT-13910-C allele in Chilean IBD patients (0.79) was comparable to the general population and higher than Europeans (0.49). Regarding VDR and VDD variants, in our study, the rs12785878-GG variant was associated with an increased risk of IBD (OR = 2.64, CI = 1.61–4.32; *p*-value = 0.001). Sixty-one percent of the Chilean IBD cohort have a genetic predisposition to lactose malabsorption, and a significant proportion exhibit genetic variants associated with VDD/VDR. Screening for LI and VDD is crucial in this Latin American IBD population.

## 1. Introduction

Inflammatory bowel disease (IBD) is a complex syndrome where several factors may influence the clinical course. Dietary issues such as low vitamin D levels are relevant because they relate to IBD severity, risk, and quality of life. Moreover, food intolerance, such as lactose intolerance (LI), might induce symptoms that can mimic an IBD flare. Despite these potentially relevant links with IBD, these conditions often go unnoticed because of a lack of awareness on the part of the clinicians, who may miss asking relevant questions.

LI corresponds to a clinical syndrome in which ingesting dairy products or lactose-containing food produces symptoms such as bloating, abdominal pain, nausea, flatulence, and diarrhea. LI is associated with lactose malabsorption, in which there is a failure of the small bowel to absorb ingested lactose because of lactase deficiency [1].

While various conditions such as IBD, celiac disease, and gut infections (among others) can cause secondary lactose malabsorption, where the malabsorption occurs because of the underlying condition [2], the most common cause of lactose malabsorption is a genetic condition called primary hypolactasia or lactase non-persistence (LNP). This condition is characterized by reduced lactase enzyme activity in the intestinal brush border, leading to lactose malabsorption and LI development in adulthood [3]. Comparative genomics and population genetic studies have established that LNP is, in fact, the wild-type phenotype, while the ability to continue lactase production after weaning is determined by genetic variation. The most frequent cause of lactose tolerance is a single-nucleotide polymorphism (SNP) located 13.9 kb upstream of the lactase (LCT) gene. This SNP, known as C-13910T, resides in a regulatory element where the T-allele dominates the persistence of LCT gene expression, while the CC genotype leads to LNP [4]. Furthermore, it has been established that this allele first appeared in European human populations between 5000 and 12,000 years before the common era (BCE). The distribution of this allele in present-day populations is explained by the selective pressure imposed by the availability of cattle milk in the context of limited access to other sources of nourishment [4]. 

Accordingly, LI can affect a significant percentage of the population, ranging from 33% to 75%. Prevalence rates vary across racial and ethnic groups, with the lowest occurrence observed in Europeans and a higher prevalence found in populations of Asian, Native American, and African descent [2,5,6]. In Chile, Miquel et al. reported that the genotype LCT-13910-CC is frequently found in the Chilean population, both in the Hispanic population (Mestizos; admixture of Spanish and Native American) and in Amerindians (Native American), with a clear ethnic influence [7]. The same group also reported a high correlation between this LNP genotype and LI symptoms measured by a lactose hydrogen breath test in the Chilean population [8].

Specifically, for IBD patients, a systematic meta-analysis and review reported that LI depends on the population’s ancestry and cannot be associated with disease activity [9]. Nevertheless, in the case of Crohn’s disease, it could be secondary to small intestine damage [10].

Because lactose intolerance (LI) presents with abdominal discomfort resembling IBD symptoms, it is essential to ascertain how these conditions can be mistaken for each other or if they coexist. Additionally, IBD has increasing prevalence and incidence in Latin American developing countries [11]. Thus, LI could be a problem in these Latin American countries since the genotype LCT-13910-CC is prevalent in this ethnic group. LI in this group could be an important issue in IBD Latin American patients since patients with LI might avoid dairy consumption, and IBD patients are at high risk for low bone mineral density (BMD) and osteoporosis. Some reports have shown that osteoporosis could affect up to 20% of IBD patients, with the risk factors in this group being steroid therapy, malnutrition, increased proinflammatory cytokines, and a deficiency in calcium and vitamin D [12,13,14,15]. Importantly, IBD patients with LI/LNP may favor vitamin D deficiency by avoiding lactose product consumption, a main source of vitamin D. This issue could be relevant in IBD since several studies have linked low vitamin D levels with disease outcomes. Epidemiological studies have shown that vitamin D deficiency (VDD) is highly prevalent among IBD patients, and low levels of vitamin D correlate not only with a higher osteoporosis risk but also with more severe disease and higher disease activity (hospitalization, surgery risk, Clostridioides difficile infection, and anti-TNF α response) [16,17,18,19,20,21,22,23,24,25]. Deficiencies in vitamin D have been associated with increased susceptibility to immune-mediated diseases, infections, and cancer [26,27,28,29]. 

Vitamin D is a pleiotropic hormone that performs several functions, including the regulation of calcium homeostasis, immune modulation, cell differentiation, and intercellular adhesion [23]. The inactive precursor of vitamin D, cholecalciferol, undergoes two hydroxylation steps to become active. The first hydroxylation step occurs in the liver, where cholecalciferol is converted into calcidiol or 25-hydroxycholecalciferol (25(OH)D). The second hydroxylation step takes place in the kidney, where calcidiol is metabolized to calcitriol or 1-α,25-dihydroxycholecalciferol (1,25(OH)2D) [15]. Both forms of vitamin D are transported in the bloodstream by being bound to the vitamin D-binding protein (VDBP), with the active form exerting its effects by binding to the vitamin D receptor (VDR) [23]. 

Vitamin D promotes an adequate function of the innate immune response due to its immunomodulatory properties that include exerting anti-inflammatory effects; modulating T- and B-cell activation, proliferation, and differentiation; maintaining intestinal barrier integrity; and modulating the gut microbiota. These mechanisms might influence IBD development and progression [30,31]. Interestingly, it has been reported that there is an association between IBD and SNPs related to VDD and the VDR [31,32,33,34,35]. SNPs associated with vitamin D levels in IBD have been reported previously [36]. A New Zealand study analyzed the relevance of serum vitamin D levels and genotypes to CD status. They found that serum vitamin D levels were significantly lower in CD than in health, and two allele variants, rs731236-A (VDR) and rs732594-A (SCUBE3), showed a significant association with serum vitamin D in CD [32]. A meta-analysis of several studies evaluated the impact of VDR gene polymorphisms on the risk of UC and CD. Four VDR polymorphisms (rs731236, TaqI; rs1544410, BsmI; rs2228570, FokI; and rs17879735, ApaI) were examined. This meta-analysis reveals that carrying the TaqI tt genotype increases CD risk in Europeans, while the ApaI “a” allele decreases CD risk in all carriers. The FokI polymorphism confers susceptibility to UC in Asians [37]. Despite a handful of clinical studies showing that lower vitamin D levels have been associated with IBD clinical relapse, current data have failed to establish a conclusive genetic association between VDD and VDR SNPs with IBD [38]. Thus, studies to explore this link are needed.

Our study aimed to evaluate the prevalence of LCT-13910-CC (LNP) in Chilean patients with IBD, as well as the genotypes associated with VDD and VDR. We found that 61% of Chilean IBD patients have a genetic predisposition to lactose malabsorption, and a significant proportion have SNPs associated with low vitamin D levels and VDR SNPs. Screening for lactose intolerance and VDD is important in this population to prevent negative IBD outcomes and an increased risk of osteoporosis. The consideration of lactose intolerance is important for symptomatic Chilean IBD patients with inactive disease. The prevalence of the CC genotype of LCT-13910 in the Latin American population suggests that lactose intolerance may affect Latin American IBD patients.

## 2. Results

### 2.1. Characterization of the Chilean Inflammatory Bowel Disease Cohort

A total of 192 patients were genotyped, and clinical data were recovered for 184 of them. Of these patients, 138 (75%) were diagnosed with ulcerative colitis (UC), while 46 (25%) were diagnosed with Crohn’s disease (CD). The median age of the patients was 49 years (range of 17–81), with a mean age at diagnosis of 36 years (range of 7–73). Within this cohort of IBD patients, it was observed that 36% exhibited extraintestinal manifestations, over 50% had a history of IBD-related hospitalization, and only 15% had undergone bowel resection surgery. Additionally, more than 50% of the patients received thiopurine treatment, while only 17% underwent anti-TNF therapy. Based on the Montreal Classification, 35% of UC cases were classified as extensive colitis, 31% as left colitis, and 25% as proctitis, and information regarding the classification was unavailable for 9% of cases. In the CD group, only 4% received a diagnosis before age 17. The most common extension of CD was colonic (L2) involvement, accounting for 46% of cases, followed by ileocolonic (L3) extension. Upper digestive tract involvement (L4) was observed in only 5% of CD cases, while 54% exhibited perianal involvement. The predominant CD phenotype was structuring (B2), observed in 37% of cases, followed by 33% exhibiting an inflammatory phenotype (B1).

#### 2.1.1. Characterization of Lactose-Malabsorption Genotypes

The genotype related to LNP, LCT-13910-CC, was found in 117 patients, while only 5 had the LCT-13910-TT genotype. Our analysis revealed that the genotype frequencies of rs4988235 (LCT-13910 genotypes) in the Chilean IBD cohort were like those of Chilean Hispanics (*p* = 0.4) but significantly different from that of Chilean Amerindians (*p* = 0.02) (see Table 1a). The LCT-13910-CC genotype was found in 61% (117/192) of the Chilean IBD cohort, similar to the prevalence observed in the Chilean Hispanic population (57.5%, 126/219). However, it was lower than the prevalence reported by Miquel in the Chilean Amerindian population, which was 88.4% (38/43) [7].

We compared the Chilean IBD cohort frequencies of rs4988235 (LCT-13910 genotypes) with those of other populations reported on Ensembl.org. Our Chilean IBD cohort was comparable to American populations (defined on Ensembl.org as individuals from Colombia, California (Mexican), Peru, and Puerto Rico) (*p* = 0.08) and significantly different from the European, East Asian, and global populations (All) (see Table 1b). The frequency of the risk allele LCT-13910-C related to LNP was similar among the Chilean IBD and Chilean control cohort, and the American population and was higher than that of the European population, as shown in Figure 1. Table 2 shows the frequency of SNPs associated with lactose intolerance, vitamin D deficiency, and the vitamin D receptor in Chilean IBD patients. 

#### 2.1.2. Characterization of Vitamin D-Deficiency-Associated SNPs

A total of 29 genetic variants associated with VDD were identified in the GWAS catalog [31]. These variants included rs12785878, rs2282679, rs10741657, rs7129781, rs4944958, rs964184, rs10859995, rs1532085, rs1800588, rs3755322, rs6600893, rs2205262, rs11723621, rs11023332, rs7041, rs12803256, rs3831470, rs78359207, rs55715230, rs306141, rs17382663, rs55791371, rs10426201, rs3750297, rs12123821, rs4845491, rs8123293, rs17217119, and rs58788626. Appendix A contains these results. Of these variants, only three (rs228679, rs12785878, and rs10741657) were available among the genotypes of the Chilean cohort.

The frequencies of genotypes associated with VDD risk were as follows: rs2282679-AA, 117 (61%); rs12785878-TT, 67(34.9%); and rs10741657-AA, 20 (10.4%). Five patients simultaneously had all three VDD risk genotypes (rs2282679-AA, rs12785878-TT, and rs10741657-AA), and thirteen patients had rs10741657-AA and rs2282679-AA. Forty patients (20%) had rs2282679-AA and rs12785878-TT simultaneously. 

We compared the frequencies of VDD risk variants in our Chilean IBD cohort with those reported in other populations. The prevalence of the rs12785878-TT risk genotype in our Chilean IBD cohort was 34.9% (67 out of 192 patients), significantly different from the frequencies observed in other populations, as shown in Table 3a. The prevalence of the rs12785878-TT risk genotype in our Chilean cohort was lower than that observed in the European population (49.9% or 251 out of 503 individuals). However, it was higher than the prevalence observed in the American population (21.9% or 76 out of 347 individuals) and East Asian population (13.7% or 69 out of 504). No significant differences were found in the distribution of genotypes for rs2282679 and rs10741657 when comparing our IBD cohort to other groups.

As mentioned in Section 4, we did not have access to data from a control Chilean group in our study. However, we utilized American genotype frequencies from the Ensembl.org database, which represents the Latin general population, to compare the distribution of genotypes in our IBD study population. This allowed us to assess the likelihood of an association between genotype and IBD risk. The rs12785878-GG genotype (associated with vitamin D levels) showed a significant association with a higher IBD risk when compared to the American population (OR = 2.64, CI = 1.61–4.32; *p*-value = 0.001), as shown in Table 3b. Moreover, we compared the genotypes distribution of rs12785878 of our IBD Chilean cohort with an IBD-US population [36], finding significant differences in their distribution.

#### 2.1.3. Characterization of Vitamin D Receptor SNPs

A total of eleven SNPs linked to the VDR and previously studied in IBD patients were specifically examined in our Chilean IBD cohort: rs1544410, rs17879735, rs731236, rs757343, rs11568820, rs7975232, rs7109294, rs10896349, rs732594, rs2981, and rs2980 [32,38,39]. Of these SNPs, those available were rs1544410, rs11568820, and rs7975232. 

The frequencies of SNPs related to the VDR were rs1544410-AA, 10 (5.2%); rs11568820-AA, 5 (2.6%); and rs7975232-AA, 46 (23.9%). Only two patients simultaneously had rs1544410-AA, rs11568820-AA, and rs7975232-AA. Ten patients simultaneously had rs1544410-AA and rs7975232-AA.

We observed significant genotype differences in the SNPs of rs1544410, rs7975232, and rs11568820, which are associated with the VDR and IBD, between our Chilean IBD cohort and the global (All), European, and East Asian populations, as shown in Table 4, Table 5 and Table 6. However, no significant differences were found when comparing our cohort with the American population. Moreover, there were no significant differences in the risk of developing IBD when we compared the frequencies of these SNPs in our IBD cohort with those of the American population as a control. 

We observed that the genotypes rs2282679-AA and rs7975232-AA, associated with vitamin D deficiency (VDD) and the vitamin D receptor (VDR), respectively, were the most prevalent in our study. Of the 192 individuals examined, 28 out of 192 (14.5%) possessed both genotypes. Additionally, we found that the combination of rs12785878-TT and rs7975232-AA was present in 17 out of the 192 individuals (8.8%).

Some patients exhibited a combination of genetic variants associated with both LNP and VDD risk. Specifically, the LCT-13910-CC genotype (associated with LNP) was found in combination with rs10741657-AA (associated with VDD) in sixteen patients (8.3%) and with rs12785878-TT (associated with VDD) in thirty-eight patients (19.8%). We also assessed the association between the LCT-13910-CC variant (LNP) and VDR risk variants and found that in twenty-nine patients (15.10%), LCT-13910-CC was present in combination with rs7975232-AA (associated with the VDR).

Furthermore, we conducted a comparative analysis of the VDR gene polymorphism distribution within our IBD cohort, comparing it with other IBD cohorts, namely, O’Sullivan’s Irish Cohort, Bentley’s New Zealand Cohort, Abraham’s USA Cohort, and Zheng’s China Cohort [33,35,38,39]. The results of this analysis are presented in Table 7.

## 3. Discussion

IBD affects nearly 3.1 million adults in the United States (US). There is a concerning trend of increasing incidence and prevalence of IBD among Hispanic and Latin American populations in both the US and Latin countries; however, knowledge of IBD in this ethnic group is scarce [40]. Thus, studies exploring clinical and genetic aspects in these understudied communities are essential to improve and personalize the disease’s diagnosis, treatment, and management. In our study, we contribute to reporting on the prevalence of genetic variants related to LNP, VDD, and VDR in a Latin IBD cohort.

The clinical importance of genetic tests for LI is debatable because the lactase activity decreases progressively over time, and the genotype analysis is not enough to diagnose LI [41,42]. We did not establish a correlation between LNP genotypes and diagnostic tests such as lactose breath tests. However, previous studies on the Chilean population have demonstrated a strong concordance between lactose breath tests and genetic tests [7,8]. Thus, in our Chilean IBD cohort, we consider the risk genotype of LNP to be meaningful.

The prevalence of genetic NLP in IBD patients varies among populations. European studies have revealed a low prevalence of the LCT-13910-CC genotype in healthy individuals and those diagnosed with IBD. A specific study on 165 IBD patients showed that only 7.3% (12 out of 165 patients) possessed the NLP genotype, and all exhibited lactose sensitivity [9]. These findings suggest that genetic lactase non-persistence (NLP) is not prevalent in this population. Conversely, there is a high prevalence of LI/LNP in the Latin American IBD population, such as in Chile, where the prevalence of genetic LNP is higher than in other groups (61% of LNP risk genotype; LCT-13910-CC). In this specific population, identifying other causes of intestinal symptoms in IBD patients not related to disease activity (such as LI/LNP) may result in important implications. Indeed, about one-third of patients with UC and a half of those with CD in remission met the Rome III criteria for irritable bowel syndrome [43]. Identifying and excluding food intolerances is a key strategy for managing overlapping symptoms of IBD and IBS [44]. Lactose intolerance, the most prevalent food intolerance worldwide and in Latin populations, often due to a high LNP genetic prevalence, must be considered as a reason for the persistence of symptoms despite remission in IBD Chilean and Latin patients.

Another important aspects to consider in this IBD group is that people with LI often avoid dairy products and milk, limiting the essential nutrients obtained through their diet, such as calcium, potassium, vitamin D, vitamin B, and protein, since milk products provide more than half of the dietary reference calcium intake [45]. Thus, the supplementation of nutrients to prevent future complications, such as osteoporosis and malnutrition, could be needed in this group of patients. For IBD and LI/LNP patients, several recommendations may be useful, such as drinking lactose-free dairy or using oral lactase supplements in certain situations. Choosing dairy products with low lactose content—such as kefir, cheese, and yogurt, or plant-based milk alternatives, such as rice or coconut milk, that are supplemented with calcium and vitamin D in an amount equivalent to or higher than that of cow’s milk—is also recommended. Additionally, considering vegetables as a source of calcium can be beneficial [45,46].

VDD is prevalent in IBD, and lower vitamin D levels are linked to osteopenia and osteoporosis, and also linked to disease activity, a higher postoperative recurrence, more frequent relapses, poorer quality of life, and the failure of response to biologics, as compared to IBD patients with normal or high levels of vitamin D [24]. This role of vitamin D in the disease course could be explained by recent research studies’ findings that show the crucial role of vitamin D in regulating innate and adaptive immunity. Low VDR levels are associated with chronic inflammation and the downregulated expression of ATG16L1 [47,48]. ATG16L1 is a gene necessary for autophagy and maintaining intestinal homeostasis. ATG16L1 affects not only the innate but also the adaptative response through its expression in the gut epithelium, dendritic cells, and T and B cells, and recently, the VDR was shown to regulate this gene transcriptionally [47,48]. Moreover, the gut epithelial lining is a vital physical barrier that requires tight and adherent junctions between epithelial cells to maintain adequate intestinal permeability. The loss of this barrier can lead to gut inflammation. In individuals with IBD, an increased expression of claudin-2, a protein that increases intestinal permeability, has been observed, and it is stimulated by interferon [IFN]-γ. In contrast, protein tyrosine phosphatase N2 (PTPN2) inhibits claudin-2 expression. The vitamin D–VDR complex plays a crucial role in maintaining the integrity of the intestinal barrier by inducing transcription of the gene coding for PTPN2, inhibiting claudin-2 expression, protecting the intestinal barrier, and maintaining the differentiated adhesive phenotype of intestinal epithelial cells. This mechanism helps to prevent gut inflammation [49,50,51]. Another important effect of vitamin D on the epithelial barrier levels occurs when the intestinal epithelium and macrophages are activated by pathogens, as they produce antimicrobial peptides such as cathelicidin and defensins to protect the intestinal barrier. Vitamin D can increase cathelicidin levels in macrophages by interacting with vitamin D response elements in the promoter region of the cathelicidin gene [52]. Low levels of 25[OH]D are associated with reduced cathelicidin expression, highlighting the importance of vitamin D in enhancing innate immune defenses [52]. Vitamin D also plays an important role in regulating innate immunity by stimulating the expression of the NOD2/CARD15/IBD1 gene and protein in epithelial and monocyte cells. This leads to the expression of antimicrobial defense beta2 and antimicrobial cathelicidin in the presence of the muramyl peptide. However, this effect is only observed in individuals with functional NOD2, as patients with Crohn’s disease who are homozygous for non-functional NOD2 variants do not display this response. Vitamin D deficiency may be linked to the genetics of Crohn’s disease, as its pathogenesis is associated with attenuated NOD2 or an antimicrobial peptide function [53]. Vitamin D might also downregulate the IL-23 receptor pathway in innate lymphoid cells (ILC-3), which are tissue-resident lymphocytes functionally resembling T-helper 17/22 cells in the adaptive system [54].

In terms of the adaptive immune response, vitamin D stimulates dendritic cells, resulting in a reduced response to lipopolysaccharides and decreased T-cell activation [55,56]. Vitamin D promotes IL-10 production by dendritic cells, leading to an anti-inflammatory state and promoting the T-helper 2 response [57]. Vitamin D also stimulates CD4+ T cells to produce IL-10 [58] and promotes the maturation of Th2 cells by increasing the transcription factors c-maf and GATA3 [24,59]. Additionally, circulating B cells can regulate the immune response by producing vitamin D through an autocrine mechanism [60].

Considering the significant associations between VDD, the VDR, and IBD, we aimed to identify SNPs related to VDD and the VDR in the Chilean IBD cohort. We found three genetic risk variants related to VDD: rs228679, rs12785878, and rs10741657. According to our analysis, rs12785978 was the only SNP related to VDD and associated with IBD risk, as shown in Table 3. The rs12785978 variant is located at chromosome 11:71456403 and maps the DHCR7 gene that encodes the enzyme 7-dehydrocholesterol reductase (DHCR7), which converts 7-dehydrocholesterol (7DHC) to cholesterol. However, 7-dehydrocholesterol is also a precursor for vitamin D synthesis when exposed to UV light. DHCR7’s enzymatic activity loss results in 7DHC accumulation, increasing vitamin D production. DHCR7 acts as a crucial regulatory switch between cholesterol and vitamin D synthesis [61,62].

The VDR gene is mapped to a region on chromosome 12 that is linked to IBD [34]. The VDR is the cellular receptor for 1,25(OH)_2_ vitamin D3 (calcitriol), which has a wide range of divergent regulatory effects on the immune system. SNPs in the VDR have been shown to alter 1,25(OH)2D3-VDR interactions [63,64], and there are several SNPs in the VDR linked to IBD [33,35,38,39].

Among the eleven SNPs related to VDR analyzed in our Chilean IBD cohort, we identified three (rs1544410, rs11568820, and rs7975232) that had significant differences in their distribution compared to other non-American populations. However, based on our analysis, these SNPs were not associated with the risk of IBD (Table 4b, Table 5b and Table 6b). We observed a significant difference in the distribution of genotypes of VDR SNP between our IBD population (Table 7) and other IBD groups. However, it is essential to note that these differences may be attributed to variances in ancestry among the studied populations rather than specific clinical factors.

The findings of this study could hold significant implications for the Chilean population, as well as other Hispanic and Amerindian populations that share a similar genetic background. However, further research is necessary in order to validate these findings and establish more robust evidence. It is important to consider the high prevalence of lactase non-persistence in these populations when managing IBD in Latin patients. Furthermore, our study revealed a notable occurrence of VDD risk genotypes, specifically rs2282679-AA, 117 (61%), and indicated that the rs12785878 genotype (associated with vitamin D levels) may serve as a risk genotype for IBD in the Chilean population, as per our analysis. These findings are intriguing and suggest that rs12785878 may have a distinct role in the susceptibility of Chileans to IBD compared to other populations.

One limitation of the study is the lack of information regarding the vitamin D levels in the study population. This study does not provide data on the participants’ vitamin D status, which could have provided additional insights into the relationship between vitamin D deficiency and IBD in the Hispanic and Amerindian populations.

It would be desirable to conduct a separate analysis based on the disease type (CD or UC), considering the distinct clinical phenotypes of UC and CD. UC is limited to the colon, while Crohn’s disease can affect the entire gastrointestinal system, including the small bowel, potentially impacting vitamin D absorption. However, due to the smaller size of the CD group, which comprised only 46 patients, we did not perform a subgroup analysis. This decision was made due to the likelihood of failing to identify true differences or relationships between variables, even if they exist in the population (type II error).

Our study contributes to understanding the prevalence of genetic variants related to lactase non-persistence and vitamin D deficiency in the Chilean IBD cohort, highlighting its potential implications for disease management and identifying other causes of symptoms. Vitamin D plays a crucial role in regulating both innate and adaptive immunity, as it affects various immune cells and pathways involved in gut inflammation and the maintenance of intestinal homeostasis. Genetic variants related to VDD and the VDR were identified in the Chilean IBD cohort, showing associations with IBD risk and differences compared to other populations. These findings have broader implications for similar populations in the Americas and underscore the importance of tailored approaches for diagnosis, treatment, and public health strategies for IBD and other related conditions.

## 4. Materials and Methods

### 4.1. Study Group

We performed an observational study in a Chilean IBD tertiary referral center, the Hospital San Borja Arriarán (HSBA), from January 2020 to December 2021. Patients were invited to participate if they had received an IBD diagnosis supported by clinical, endoscopic, histologic, and imaging findings according to clinical guidelines and International Disease Classification criteria [65,66,67]. Ethics approval was given by the Institutional Review Boards of Servicio de Salud Metropolitano Central/HSBA (IRB:43/2022). All patients provided written informed consent.

### 4.2. Genotyping

In total, 5 mL of blood was collected from each participant and stored in plastic vacutainer tubes containing ethylenediaminetetraacetic (EDTA). DNA from peripheral blood was extracted using the Invisorb Blood Universal (Invitek) # ref 1031150200 purification kit, following the manufacturer’s instructions. Samples were stored at −80 °C until genotyped at Erasmus MC-Netherlands using Illumina’s Infinium Global Screening Array, and a total of 725.497 SNPs were investigated.

### 4.3. Data Analysis

The data were read from binary (bed) and index (bim) files, and a subset of variants were filtered based on a list of variants of interest. The resulting genotypes were then transformed from the wide format to the long format, merged with a bim file to add the variant information, and transformed back to the wide format. The final dataset was saved in the CSV format for further analysis. The R libraries used were “genio”, “plinkFile”, “readr”, and “tidyverse”.

We examined the allele frequencies of LCT-13910 (rs4988235) in the genotyped cohort of IBD patients in Chile. Additionally, we obtained a list of SNPs related to VDD from the GWAS Catalog [24]. We further investigated twelve SNPs that are associated with both IBD and the vitamin D receptor (VDR) [27,28,29,62]. These SNPs were looked for among the 725.497 SNPs investigated using the R programming language.

We first compared the frequency of the LCT-13910 genotypes of the Chilean IBD cohort with the genetic frequencies reported for Chilean Hispanics and Amerindians by Miquel et al. [9]. Subsequently, we conducted a comparative analysis of risk allele frequencies and genotype frequencies for variants of interest related to LNP and VDD in our IBD cohort from Chile, comparing the data with those reported for global (All), American, East Asian, and European populations on Ensembl.org. The “All” category on the website “Ensembl.org” represents African, American, East Asian, European, and South Asian populations. The American population is further represented by individuals from Colombia, California (Mexican), Peru, and Puerto Rico. The East Asian population consists of individuals from China, Japan, and Vietnam, while the European population includes individuals from Finland, Britain, Iberia, and Italy, as well as individuals from Utah with northern and western European ancestry.

We conducted a chi-square test with a significance level of 0.05 to assess potential associations between these cohorts. Furthermore, we calculated the odds ratio (OR) and confidence interval (CI) for the SNPs associated with VDD and the VDR. Since we did not have a Chilean control group available, we utilized the data published on Ensembl.org as controls. This dataset pertains to the American population and includes Latin individuals, making it potentially representative of the Chilean population. It is important to note that the proportion of individuals with IBD in this control sample should reflect the corresponding proportions in the general population. Consequently, we would only expect a small number of affected individuals within the control group.

Additionally, we compared the genotype distribution of rs1278585878 (associated with VDD) and the VDR SNPs identified in our population with those of other IBD cohorts. The genotypes and risk allele frequency were obtained from previous studies [33,35,38,39]

We used the R packages epitools, readxl, and rapportools and the functions chisq.test() and oddsratio.wald() to analyze genetic data and provide statistical insights through chi-squared testing and odds ratio calculation.

## Figures and Tables

**Figure 1 ijms-24-14866-f001:**
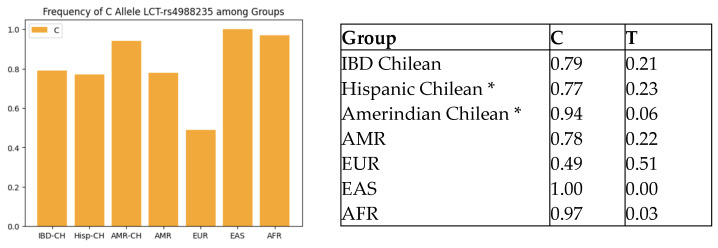
Comparison of LCT-rs4988235-C allele frequency across populations. IBD-CH = IBD Chilean; Hisp-CH = Hispanic Chilean; AMR-CH = Amerindian Chilean; EUR = European; EAS = East Asian; AFR = African. * Miquel et al. [9].

**Table 1 ijms-24-14866-t001:** Genotype frequencies of rs4988235 (LCT-13910) in IBD Chilean cohort and comparative analysis with Chilean controls. (**a**) Comparison between Chilean cohorts. (**b**) Comparison between the Chilean IBD cohort and other populations reported on Ensembl.org.

LCT-13910 rs4988235	TT	TC	CC	*p*-Value (Chi-Square)
Chilean IBD (N = 192)	5 (2.6%)	70 (36.4%)	117 (61%)	0.4
Chilean Hispanic * (N = 219)	3 (1.4%)	90 (41.1%)	126 (57.5%)	Reference
Amerindians * (N = 43)	0	5 (2.6%)	38 (88.4%)	0.02
(**a**)				
LCT-13910 rs4988235	TT	TC	CC	*p*-Value (Chi-square)
Chilean IBD (N = 192)	5 (2.6%)	70 (36.4%)	117 (61%)	Reference
All (N = 2504)	197 (7.9%)	414 (16.5%)	1893 (75.6%)	7.74 × 10^−12^
European (N = 503)	162 (32.2%)	187 (37.2%)	154 (30.6%)	<2.2 × 10^−6^
American (N = 347)	22 (63%)	106 (30.6%)	219 (63.1%)	0.08
East Asian (N = 504)	0	0	504 (100%)	<2.2 × 10^−6^
(**b**)				

* Data obtained from Miquel et al. [9].

**Table 2 ijms-24-14866-t002:** Single-nucleotide polymorphisms (SNPs) associated with lactose intolerance, vitamin D deficiency, and the vitamin D receptor in Chilean IBD patients. LCT = gene that encodes for the lactase enzyme; GC = gene that encodes the human-group-specific component, which is the major vitamin-D-binding protein (VDBP) in plasma; DHCR7 = gene that produces the 7-dehydrocholesterol reductase enzyme, which catalyzes the production of cholesterol from 7-dehydrocholesterol using NADPH. The CYP2R1 gene has been linked by several studies to vitamin D serum concentrations.

Risk Variant	Chr	Gene	Risk Allele	Description	Allele Frequency		Genotype		
				Encodes for the lactase enzyme	C = 0.79		CC	TC	TT
rs4988235	2	LCT	C		T = 0.21	Lactose malabsorption	117 (61%)	70 (36.4%)	5 (2.6%)
				Encodes human-group-specific component, which is the major vitamin D-binding protein (VDBP) in plasma	C = 0.22 A = 0.88	Deficiency of vitamin D	CC	CA	AA
rs2282679	4	GC	A				10 (5.2%)	65 (33.8%)	117 (61%)
				Gene that produces the 7-dehydrocholesterol reductase enzyme, which catalyzes the production of cholesterol from 7-dehydrocholesterol using NADPH	G = 0.42 T = 0.58		GG	GT	TT
rs12785878	11	DHCR7	T			Deficiency of vitamin D	37 (19.3%)	88 (45.8%)	67 (34.9%)
				CYP2R1 gene has been linked by several studies to vitamin D serum concentrations	G = 0.67		GG	AG	AA
rs10741657	11	CYP2R1	A		A = 0.33	Deficiency of vitamin D	86 (44.8%)	86 (44.8%)	20 (10.4%)
				VDR (vitamin D receptor) gene variants	G = 0.758 A = 0.242		AA	AG	GG
rs1544410	12	VDR	A			Related to autoimmune or inflammatory disease	10 (5.2%)	73 (38%)	109 (56.78%)
				VDR (vitamin D receptor) gene variants	G = 0.872 A = 0.127	Related to autoimmune or inflammatory disease	AA	AG	GG
rs11568820	12	VDR	A				5 (2.60%)	39 (20.31%)	148 (77.08%)
				VDR (vitamin D receptor) gene variants	C = 0.531 A = 0.469	Related to autoimmune or inflammatory disease	AA	AC	CC
rs7975232	12	VDR	A				46 (23.96%)	88 (45.83%)	58 (30.21)

**Table 3 ijms-24-14866-t003:** Single-nucleotide polymorphism (SNP) rs12785878 associated with vitamin D deficiency: comparison between Chilean IBD group and other (**a**) populations reported on Ensembl.org; (**b**) IBD risk estimation for rs12785878; (**c**) IBD-US population [36].

(**a**)											
rs12785878			Chilean IBD (N = 192)			All (N = 2504)			*p*-Value		
TT			67 (34.9)			433 (17.3%)			1.32 × 10^−14^		
GT			88 (45.8%)			907 (36.2%)					
GG			37 (19.3%)			1164 (46.5%)					
rs12785878			Chilean IBD (N = 192)			European (N = 503)			*p*-value		
TT			67 (34.9)			251 (49.9%)			0.0001		
GT			88 (45.8%)			203 (40.4%)					
GG			37 (19.3%)			49 (9.7%)					
rs12785878			Chilean IBD (N = 192)			American (N = 347)			*p*-value		
TT			67 (34.9)			76 (21.9%)			0.0005		
GT			88 (45.8%)			160 (46.1%)					
GG			37 (19.3%)			111 (32%)					
rs12785878			Chilean IBD (N = 192)			East Asian (N = 504)			*p*-value		
TT			67 (34.9)			69 (13.7%)			7.24 × 10^−11^		
GT			88 (45.8%)			245 (48.6%)					
GG			37 (19.3%)			190 (37.7%)					
(**b**)											
rs12785878		Chilean IBD (N = 192)		American (N = 347)		OR		CI		*p*-Value	
TT		67 (34.9)		76 (21.9%)		Reference		Reference		Reference	
GT		88 (45.8%)		160 (46.1%)		1.60		1.05–2.43		0.03	
GG		37 (19.3%)		111 (32%)		2.64		1.61–4.32		0.0001	
(**c**)											
rs12785878			Chilean IBD N = 192			Xavier N = 478			*p* Value (Chi^2^)		
TT			67 (34.9%)			235 (49%)			0.001		
GT			88 (45.8%)			186 (39%)					
GG			37 (19.39%)			57 (12%)					
T vs. G			0.57			0.68					

OR = odds ratio; CI = confidence interval.

**Table 4 ijms-24-14866-t004:** Comparison of single-nucleotide polymorphism rs1544410, associated with the vitamin D receptor, between the Chilean IBD group and other populations reported on Ensembl.org. (**a**) Comparison frequencies of genotypes of rs1544410 between populations. (**b**) IBD risk estimation for the SNP rs1544410.

(**a**)											
rs1544410			Chilean IBD (N = 192)			All (N = 2504)			*p*-Value		
GG			109 (56.8%)			1303 (52%)			0.03		
AG			73 (38%)			920 (36.8%)					
AA			10 (5.2%)			281 (11.2%)					
rs1544410			Chilean IBD (N = 192)			European (N = 503)			*p*-Value		
GG			109 (56.8%)			186 (37%)			3.47 × 10^−7^		
AG			73 (38%)			228 (45.3%)					
AA			10 (5.2%)			89 (17.7%)					
rs1544410			Chilean IBD (N = 192)			American			*p*-Value		
GG			109 (56.8%)			196 (56.2%)			0.5		
AG			73 (38%)			124 (35.7%)					
AA			10 (5.2%)			27 (7.8%)					
rs1544410			Chilean IBD (N = 192)			East Asian			*p*-Value		
GG			109 (56.8%)			439 (87.1%)			<2.2 × 10^−16^		
AG			73 (38%)			65 (12.9%)					
AA			10 (5.2%)			0					
(**b**)											
rs1544410		Chilean IBD (N = 192)		American		OR		CI		*p*-Value	
GG		109 (56.8%)		196 (56.2%)		Reference		Reference		Reference	
AG		73 (38%)		124 (35.7%)		0.94		0.65–1.37		0.76	
AA		10 (5.2%)		27 (7.8%)		1.50		0.70–3.21		0.29	

OR = odd ratio; CI = confidence interval.

**Table 5 ijms-24-14866-t005:** Comparison of single-nucleotide polymorphism rs7975232, associated with the vitamin D receptor, between the Chilean IBD group and other populations reported on Ensembl.org. (**a**) Comparison frequencies of genotypes of rs7975232 between populations. (**b**) IBD risk estimation for rs7975232.

(**a**)											
rs7975232			Chilean IBD (N = 192)			All (N = 2504)			*p*-Value		
CC			58 (30%)			657 (26.2%)			0.24		
AA			46 (24%)			734 (29.3%)					
AC			88 (46%)			1113 (44.4%)					
rs7975232			Chilean IBD (N = 192)			European (N = 503)			*p*-Value		
CC			58 (30%)			115 (22.9%))			0.02		
AA			46 (34%)			170 (33.8%)					
AC			88 (46%)			218 (43.3%)					
rs7975232			Chilean IBD (N = 192)			American (N = 347)			*p*-Value		
CC			58 (30%)			116 (33.4%)			0.69		
AA			46 (34%)			75 (21.6%)					
AC			88 (46%)			156 (45%)					
rs7975232			Chilean IBD (N = 192)			Asian (N = 504)			*p*-Value		
CC			58 (30%)			261 (51.8%)			1.56 × 10^−8^		
AA			46 (34%)			50 (9.9%)					
AC			88 (46%)			193 (38.3%)					
(**b**)											
rs7975232		Chilean IBD (N = 192)		American (N = 347)		OR		CI		*p*-Value	
CC		58		116 (33.4%)		Reference		Reference		Reference	
AA		46		75 (21.6%)		0.81		0.50–1.32		0.40	
AC		88		156 (45%)		0.88		0.58–1.33		0.56	

OR = odds ratio; CI = confidence interval.

**Table 6 ijms-24-14866-t006:** Comparison of single-nucleotide polymorphism (SNP) rs11568820, associated with the vitamin D receptor, between the Chilean IBD group and other populations reported on Ensembl.org. (**a**) Comparison frequencies of genotypes of rs11568820 between populations. (**b**) IBD risk estimation for the SNP rs11568820.

(**a**)											
rs11568820			Chilean IBD (N = 192)			All (N = 2504)			*p*-Value		
GG			148 (77.1%)			941 (37.6%)			<2.2 × 10^−16^		
GA			39 (20.3%)			838 (33.5%)					
AA			5 (2.6%)			725 (29%)					
rs11568820			Chilean IBD (N = 192)			European (N = 503)			*p*-Value		
GG			148 (77.1%)			303 (60.2%)			0.0001		
GA			39 (20.3%)			171 (34%)					
AA			5 (2.6%)			29 (5.8%)					
rs11568820			Chilean IBD (N = 192)			East Asian (N = 504)			*p*-Value		
GG			148 (77.1%)			189 (37.5%)			<2.2 × 10^−16^		
GA			39 (20.3%)			231 (45.8%)					
AA			5 (2.6%)			84 (16.7%)					
rs11568820			Chilean IBD (N = 192)			American (N = 347)			*p*-Value		
GG			148 (77.1%)			235 (67.7%)			0.06		
GA			39 (20.3%)			97 (28%)					
AA			5 (2.6%)			15 (4.3%)					
(**b**)											
rs11568820		Chilean IBD (N = 192)		American (N = 347)		OR		CI		p-Value	
GG		148 (77.1%)		235 (67.7%)		Reference		Reference		Reference	
GA		39 (20.3%)		97 (28%)		1.56		1.02–2.39		0.03	
AA		5 (2.6%)		15 (4.3%)		1.88		0.67–5.30		0.22	

OR = odds ratio; CI = confidence interval.

**Table 7 ijms-24-14866-t007:** Comparison of VDR single-nucleotide polymorphisms in IBD cohorts.

SNP	Inflammatory Bowel Disease Group Study			
rs1544410 (Bsml)	Chilean IBD N = 192	0’Sullivan N = 645	Abraham N = 50	Zheng N = 404
GG	109 (57%)	216 (33.5%)	9 (18%)	366 (90.6%)
GA	73 (38%)	312 (48.4%)	24 (48%)	38 (9.4%)
AA	10 (5%)	117 (18.1%)	17 (34%)	0
G vs. A	0.76	0.42	0.42	0.95
rs7975232 (Apal)	Chilean IBD N = 192	0′Sullivan N = 641	Zheng N = 404	
AA	46 (34%)	205 (32%)	196 (48.5%)	
CA	88 (46%)	306 (47.7%)	185 (45.8%)	
CC	58 (30%)	130 (20.3%)	23 (5.7%)	
A vs. C	0.47	0.44	0.29	
rs11568820	Chilean IBD N = 192	Bentley N = 897		
AA	5 (2.6%)	44 (5% )		
AG	39 (20.3%)	308 (34%)		
GG	148 (77.1%)	545 (61%)		
A vs. G	0.13	0.22		

Data obtained from Zheng [33], O′Sullivan [35], Abraham [38], and Bentley’s [39] studies.

## Data Availability

This study received approval from the Ethics Committee, which states that the authors are not permitted to share the raw data in public repositories. However, data sharing is possible through academic collaborations. Researchers who wish to access the data can contact the corresponding authors directly.

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
