# Peer review of "Genotype Prevalence of Lactose Deficiency, Vitamin D Deficiency, and the Vitamin D Receptor in a Chilean Inflammatory Bowel Disease Cohort: Insights from an Observational Study"

_ijms, 2023, doi:10.3390/ijms241914866_

Round 1

Reviewer 1 Report

l  The method section of the abstract should be revised and rewritten because it was basically written too imprecisely. Please supplement with content on the study population and research methodology.

l  The Introduction section is so disorganized that it seems unfocused. The author should rewrite and highlight the association between IBD, lactase non-persistence and vitamin D deficiency.

l  Figure 1 in your paper is a bit blurry. Please consider replacing it with a clearer one.

l  UC and CD are two clinical phenotypes of IBD. Left-sided colitis is the commonest disease location in UC, while CD involves the entire gastrointestinal system from the oral cavity to the anus, especially the small intestine, where Vitamin D is absorbed. Therefore, I recommend that the authors perform subgroup analyses for UC and CD.

l  The authors only compared their observations with published data from different studies in the general population, and the application of publicly available datasets of ethnically diverse IBD populations could validate the results and bring more reliable conclusions.

l  The method section of the abstract should be revised and rewritten because it was basically written too imprecisely. Please supplement with content on the study population and research methodology.

l  The Introduction section is so disorganized that it seems unfocused. The author should rewrite and highlight the association between IBD, lactase non-persistence and vitamin D deficiency.

l  Figure 1 in your paper is a bit blurry. Please consider replacing it with a clearer one.

l  UC and CD are two clinical phenotypes of IBD. Left-sided colitis is the commonest disease location in UC, while CD involves the entire gastrointestinal system from the oral cavity to the anus, especially the small intestine, where Vitamin D is absorbed. Therefore, I recommend that the authors perform subgroup analyses for UC and CD.

l  The authors only compared their observations with published data from different studies in the general population, and the application of publicly available datasets of ethnically diverse IBD populations could validate the results and bring more reliable conclusions.

Author Response

Dear Editor, and Reviewers

Thanks very much for your comments on our Manuscript ID: ijms-2612415, Tittle: Genotype Prevalence of Lactose Intolerance, Vitamin D Deficiency, and Vitamin D Receptor in a Chilean Inflammatory Bowel Disease Cohort: Insights from an Observational Study. Thanks again for the time and effort dedicated to improving our manuscript.

Regarding English language, we attached the certificated that this article was edited by MDPI, previously to submitted for first time. In addition, MDPI give guarantee to edit and correct again the English. They mentioned “Please note that this manuscript did not require an extensive level of editing, as the language in both the previously edited and newly edited sections were of a high standard and did not require much intervention”. In this case, the language in your manuscript is clear, consistent, grammatically correct, and suitable for submission.

We checked and added more references relevant to the contents of the manuscript.

We include the suggestions of the reviewers as is explained below.

All the text changes were highlighted in yellow.

We could not address only one comments from the reviewer 1, we explained the reason below, and add this commentary in the discussion.

We hope the new version meets your expectations.

Yours

sincerely,
The authors

We copied the answers below, and we also attached a file with the replies.

  1. The method section of the abstract should be revised and rewritten because it was basically written too imprecisely. Please supplement with content on the study population and research methodology.

Thanks. We improved this section. We added more information to the methods in the abstracts. Lines 28 to 33.

(2) Methods: An observational study in a Chilean IBD cohort was conducted. Blood samples were collected and genotyped using Illumina's Infinium Global Screening Array, analyzing 725,497 SNPs. Genetic variants related to LI, VDD, and VDR were examined in 192 IBD patients. Comparative analyses compared our findings with other populations from public genomic databases. A chi-square test assessed the potential association between cases and controls.

  1. The Introduction section is so disorganized that it seems unfocused. The author should rewrite and highlight the association between IBD, lactase non-persistence and vitamin D deficiency.

3.      Thanks for your observation. We considered your suggestion and rewrote the introduction extensively including adding an opening paragraph to guide the readers to the aims of the manuscript, highlighted in yellow in the revised text. Other paragraphs have also been edited for clarity, see Lines 46 to 147.

Inflammatory Bowel Disease (IBD) is a complex syndrome where several factors may influence the clinical course. Dietary issues such as low vitamin D levels are relevant because they relate to IBD severity, risk, and quality of life. Moreover, food intolerance, such as lactose intolerance (LI), might induce symptoms that can mimic an IBD flare. Despite these potentially relevant links with IBD, these conditions often go unnoticed because of a lack of awareness on the part of the clinicians, who may miss asking relevant questions.

LI corresponds to a clinical syndrome….      .

                    . …Importantly, IBD patients with LI/LNP may favor vitamin D deficiency by avoiding lactose product consumption, a main source of vitamin D. This issue could be relevant in IBD since several studies have linked low vitamin D levels with disease outcomes. Epidemiological studies have shown that vitamin D deficiency (VDD) is highly prevalent among IBD patients, and low levels of vitamin D correlate not only with a higher osteoporosis risk but also with more severe disease and higher disease activity (hospitalization, surgery risk, Clostridioides difficile infection, and anti-TNF α response) [16]–[25]. Deficiencies in vitamin D have been associated with increased susceptibility to immune-mediated diseases, infections, and cancer [26]–[29].

           …These mechanisms might influence IBD development and progression [30][31]. Interestingly, it has been reported that there is an association between IBD and SNPs related to VDD and the VDR [31]–[35]. SNPs associated with vitamin D levels in IBD have been reported previously[36]. A New Zealand study analyzed the relevance of serum vitamin D levels and genotypes to CD status. They found that serum vitamin D levels were significantly lower in CD than in health, and two allele variants, rs731236-A (VDR) and rs732594-A (SCUBE3), showed a significant association with serum vitamin D in CD [32]. A meta-analysis of several studies evaluated the impact of VDR gene polymorphisms on the risk of UC and CD. Four VDR polymorphisms (rs731236; TaqI, rs1544410; BsmI, rs2228570; FokI, and rs17879735; ApaI) were examined. This meta-analysis reveals that carrying the TaqI tt genotype increases CD risk in Europeans, while the ApaI "a" allele decreases CD risk in all carriers. The FokI polymorphism confers susceptibility to UC in Asians [37]. Despite a handful of clinical studies showing that lower vitamin D levels have been associated with IBD clinical relapse, current data have failed to establish a conclusive genetic association between VDD and VDR SNPs with IBD [38]. Thus, studies to explore this link are needed.

Our study aimed to evaluate the prevalence of LCT-13910-CC (LNP) in Chilean patients with IBD, as well as the genotypes associated with VDD and VDR. We found that 61% of Chilean IBD patients have a genetic predisposition to lactose malabsorption, and a significant proportion have SNPs associated with low vitamin D levels and VDR SNPs. Screening for lactose intolerance and VDD is important in this population to prevent negative IBD outcomes and an increased risk of osteoporosis. The consideration of lactose intolerance is important for symptomatic Chilean IBD patients with inactive disease. The prevalence of the CC genotype of LCT-13910 in the Latin American population suggests that lactose intolerance may affect Latin American IBD patients.

  1. Figure 1 in your paper is a bit blurry. Please consider replacing it with a clearer one.

Thanks for the commentary the figure was changed.

  1. UC and CD are two clinical phenotypes of IBD. Left-sided colitis is the commonest disease location in UC, while CD involves the entire gastrointestinal system from the oral cavity to the anus, especially the small intestine, where Vitamin D is absorbed. Therefore, I recommend that the authors perform subgroup analyses for UC and CD.

We obtained clinical data from 184 out of the 193 genotyped patients in our study. Most of the patients had ulcerative colitis (UC) (138 patients, accounting for 75% of the sample), while a smaller proportion had Crohn's disease (CD) (46 patients, accounting for 25% of the sample). Considering the smaller sample size, especially in the case of CD, and the potential risk of type II errors, we decided to analyze both diseases together as IBD (Crohn's disease and UC). However, we acknowledge the limitations imposed by the reduced sample size, and we will address this deficiency in the discussion section of our study.

Lines 437-443

It would be desirable to conduct a separate analysis based on the disease type (CD or UC), considering the distinct clinical phenotypes of UC and CD. UC is limited to the colon, while Crohn's disease can affect the entire gastrointestinal system, including the small bowel, potentially impacting vitamin D absorption. However, due to the smaller size of the CD group, which comprised only 46 patients, we did not perform a subgroup analysis. This decision was made due to the likelihood of failing to identify true differences or relationships between variables, even if they exist in the population (type II error).

  1. The authors only compared their observations with published data from different studies in the general population, and the application of publicly available datasets of ethnically diverse IBD populations could validate the results and bring more reliable conclusions.

We Thank you for the observation. We added Tables 3c and 7, where we compared the distribution of Chilean IBD SNPs with other populations and commented on the text. Lines 234-236.

Moreover, we compared the genotypes distribution of rs12785878 of our IBD Chilean cohort with an IBD-US population [36], finding significant differences in their distribution.

Lines 295-299

Furthermore, we conducted a comparative analysis of the VDR gene polymorphism distribution within our IBD cohort, comparing it with other IBD cohorts, namely, O'Sullivan’s Irish Cohort, Bentley’s New Zealand Cohort, Abraham’s USA Cohort, and Zheng’s China Cohort[33], [35], [38], [39]. The results of this analysis are presented in Table 7.

In fact, this analysis led us to conclude that one of our results (i.e., frequencies of FokI polymorphism on VDR gene) requires further validation before being published and we have therefore removed it from our analysis.

, and Lines 418-422

We observed a significant difference in the distribution of genotypes of VDR SNP between our IBD population (Table 7) and other IBD groups. However, it is essential to note that these differences may be attributed to variances in ancestry among the studied populations rather than specific clinical factors.

Reviewer 2 Report

Dear Editors,

Thank you for the opprotunity to revise article „Genotype Prevalence of Lactose Intolerance, Vitamin D Deficiency, and the Vitamin D Receptor in a Chilean Inflammatory Bowel Disease Cohort: Insights from an Observational Study”.

The article is very interesting. Lactose intolerance and vitamin D deficiency have been associated with inflammatory bowel disease. The CC genotype of the lactase gene (LCT-13910) has been linked to lactose intolerance, whereas viamin D deficiency and the vitamin D receptor may play a role in IBD. The study assessed the prevalence of the LCT-13910 CC genotype in Chilean IBD patients and evaluated VDD and VDR genotypes.

The article is well written. The introduction section is quiet short and might be improved.

The material and methods section is very detailed and there is no need to change anything.

The results are clearly demonstarted and the discussion is rigorous. I recommend to decsribe patients more detailed (medications, treatment, were they on special diet, how were they diagnosed, etc.)

Thanks.

Author Response

Dear Editor, and Reviewers

Thanks very much for your comments on our Manuscript ID: ijms-2612415, Tittle: Genotype Prevalence of Lactose Intolerance, Vitamin D Deficiency, and Vitamin D Receptor in a Chilean Inflammatory Bowel Disease Cohort: Insights from an Observational Study. Thanks again for the time and effort dedicated to improving our manuscript.

Regarding English language, we attached the certificated that this article was edited by MDPI, previously to submitted for first time. In addition, MDPI give guarantee to edit and correct again the English. They mentioned “Please note that this manuscript did not require an extensive level of editing, as the language in both the previously edited and newly edited sections were of a high standard and did not require much intervention”. In this case, the language in your manuscript is clear, consistent, grammatically correct, and suitable for submission.

We checked and added more references relevant to the contents of the manuscript.

We include the suggestions of the reviewers as is explained below.

All the text changes were highlighted in yellow.

We could not address only one comments from the reviewer 1, we explained the reason below, and add this commentary in the discussion.

We hope the new version meets your expectations.

Yours

sincerely,
The authors

PS. We replied here and also attached a file

1.The article is well written. The introduction section is quiet short and might be improved.

1.      Thanks, for your feedback and commentary. We improve the introduction according to your suggestions.

2.           Lines 46 to 146

Inflammatory Bowel Disease (IBD) is a complex syndrome where several factors may influence the clinical course. Dietary issues such as low vitamin D levels are relevant because they relate to IBD severity, risk, and quality of life. Moreover, food intolerance, such as lactose intolerance (LI), might induce symptoms that can mimic an IBD flare. Despite these potentially relevant links with IBD, these conditions often go unnoticed because of a lack of awareness on the part of the clinicians, who may miss asking relevant questions.

LI corresponds to a clinical syndrome….      .

                    . …Importantly, IBD patients with LI/LNP may favor vitamin D deficiency by avoiding lactose product consumption, a main source of vitamin D. This issue could be relevant in IBD since several studies have linked low vitamin D levels with disease outcomes. Epidemiological studies have shown that vitamin D deficiency (VDD) is highly prevalent among IBD patients, and low levels of vitamin D correlate not only with a higher osteoporosis risk but also with more severe disease and higher disease activity (hospitalization, surgery risk, Clostridioides difficile infection, and anti-TNF α response) [16]–[25]. Deficiencies in vitamin D have been associated with increased susceptibility to immune-mediated diseases, infections, and cancer [26]–[29].

           …These mechanisms might influence IBD development and progression [30][31]. Interestingly, it has been reported that there is an association between IBD and SNPs related to VDD and the VDR [31]–[35]. SNPs associated with vitamin D levels in IBD have been reported previously[36]. A New Zealand study analyzed the relevance of serum vitamin D levels and genotypes to CD status. They found that serum vitamin D levels were significantly lower in CD than in health, and two allele variants, rs731236-A (VDR) and rs732594-A (SCUBE3), showed a significant association with serum vitamin D in CD [32]. A meta-analysis of several studies evaluated the impact of VDR gene polymorphisms on the risk of UC and CD. Four VDR polymorphisms (rs731236; TaqI, rs1544410; BsmI, rs2228570; FokI, and rs17879735; ApaI) were examined. This meta-analysis reveals that carrying the TaqI tt genotype increases CD risk in Europeans, while the ApaI "a" allele decreases CD risk in all carriers. The FokI polymorphism confers susceptibility to UC in Asians [37]. Despite a handful of clinical studies showing that lower vitamin D levels have been associated with IBD clinical relapse, current data have failed to establish a conclusive genetic association between VDD and VDR SNPs with IBD [38]. Thus, studies to explore this link are needed.

Our study aimed to evaluate the prevalence of LCT-13910-CC (LNP) in Chilean patients with IBD, as well as the genotypes associated with VDD and VDR. We found that 61% of Chilean IBD patients have a genetic predisposition to lactose malabsorption, and a significant proportion have SNPs associated with low vitamin D levels and VDR SNPs. Screening for lactose intolerance and VDD is important in this population to prevent negative IBD outcomes and an increased risk of osteoporosis. The consideration of lactose intolerance is important for symptomatic Chilean IBD patients with inactive disease. The prevalence of the CC genotype of LCT-13910 in the Latin American population suggests that lactose intolerance may affect Latin American IBD patients.

3.       

  1. The material and methods section are very detailed and there is no need to change anything.

Thanks for your commentary, we appreciate it.

  1. The results are clearly demonstarted and the discussion is rigorous. I recommend to decsribe patients more detailed (medications, treatment, were they on special diet, how were they diagnosed, etc.)

Thanks for your commentary. We added more detailed information about patients. Lines 153-165.

Within this cohort of IBD patients, it was observed that 36% exhibited extraintestinal manifestations, over 50% had a history of IBD-related hospitalization, and only 15% had undergone bowel resection surgery. Additionally, more than 50% of the patients received thiopurine treatment, while only 17% underwent anti-TNF therapy. Based on the Montreal Classification, 35% of UC cases were classified as extensive colitis, 31% as left colitis, and 25% as proctitis, and information regarding the classification was unavailable for 9% of cases. In the CD group, only 4% received a diagnosis before age 17. The most common extension of CD was colonic (L2) involvement, accounting for 46% of cases, followed by ileocolonic (L3) extension. Upper digestive tract involvement (L4) was observed in only 5% of CD cases, while 54% exhibited perianal involvement. The predominant CD phenotype was structuring (B2), observed in 37% of cases, followed by 33% exhibiting an inflammatory phenotype (B1).

The IBD diagnosis was supported by clinical, endoscopic, histologic, and imaging findings according to clinical guidelines and International Disease Classification criteria [65]–[67]. It is mentioned in the Methods

Round 2

Reviewer 1 Report

Congratulations to the author for solving the problem I raised earlier. However, the abstract still needs to be simplified.

Author Response

Dear Reviewer,
We greatly appreciate your valuable suggestions, which have significantly enhanced the quality of our article. Enclosed, you will find a more concise abstract based on your feedback. However, please do not hesitate to inform us if further revisions are necessary. We are committed to working until we meet all the publication requirements.
Thank you once again for your guidance and expertise.
Best regards,
Tamara

I copy the new abstract and I attached the new version

Abstract: (1) Background. Lactose intolerance (LI) and vitamin D deficiency (VDD) have been linked to inflammatory bowel disease (IBD). We studied the prevalence of a specific gene variant (LCT-13910 CC genotype) associated with LI and evaluated the prevalence of VDD/Vitamin D Receptor (VDR) gene variants in Chilean IBD patients. (2) Methods: We conducted an observational study in a group of Chilean IBD patients (n=192). Blood samples were collected and analyzed using Illumina's Infinium Global Screening Array to identify genetic variants related to LI, VDD, and VDR. We compared our findings with genomic data from other populations. Statistical analysis was performed to assess the association between cases and controls. (3) Results: The LCT-13910 CC genotype was found in 61% of the IBD patients, similar to the genotype frequency in Chilean Hispanic controls but lower than that in Chilean Amerindian controls. The frequency of the LCT-13910-C allele in Chilean IBD patients was 0.79, similar to the general population and higher than that in Europeans (0.49). Regarding VDR and VDD variants, in our study, the rs12785878-GG variant was associated with an increased risk of IBD (OR = 2.64, CI = 1.61-4.32; p-value = 0.001). 4) Conclusion: Sixty-one percent of Chilean IBD patients have a genetic predisposition to lactose malabsorption. A significant proportion of Chilean IBD patients exhibit genetic variants associated with VDD/VDR. Screening for LI and VDD could be crucial in this Latin American IBD population.
